# Limit Efficiency of a Silicon Betavoltaic Battery with Tritium Source

**DOI:** 10.3390/mi14112015

**Published:** 2023-10-29

**Authors:** Mykhaylo Evstigneev, Mohammad Afkani, Igor Sokolovskyi

**Affiliations:** 1Department of Physics and Physical Oceanography, Memorial University of Newfoundland, St. John’s, NL A1B 3X7, Canada; 2V. Lashkaryov Institute of Semiconductor Physics, NAS of Ukraine, 41 Prospect Nauky, 03028 Kyiv, Ukraine

**Keywords:** betavoltaic effect, energy harvesting, silicon, tritium, efficiency, radiative recombination, Auger recombination, thin-base approximation, optimization

## Abstract

An idealized design of a silicon betavoltaic battery with a tritium source is considered, in which a thin layer of tritiated silicon is sandwiched between two intrinsic silicon slabs of equal width, and the excess charge carriers are collected by thin interdigitated n^+^ and p^+^ electrodes. The opposite sides of the device are covered with a reflecting coating to trap the photons produced in radiative recombination events. Due to photon recycling, radiative recombination is almost ineffective, so the Auger mechanism dominates. An analytical expression for the current–voltage curve is obtained, from which the main characteristics of the cell, namely, the open-circuit voltage, the fill factor, and the betaconversion efficiency, are found. The analytical results are shown to agree with the numerical ones with better than 0.1% accuracy. The optimal half-thickness of this device is found to be around 1.5 μm. The maximal efficiency increases logarithmically with the surface activity of the beta-source and has the representative value of 12.07% at 0.1 mCi/cm^2^ and 14.13% at 10 mCi/cm^2^.

## 1. Introduction

Betavoltaic batteries are long-lifetime low-power sources that convert the energy of electrons produced in a beta-decay reaction into electricity [1,2,3]. They have a special niche in industrial applications, which includes autonomous devices used in hard-to-reach areas, such as outer space [4,5] or a human body, where beta-batteries can power up implants of various kinds [6].

The operation principle of a beta-battery is similar to that of a solar cell [6,7]. Beta-particles entering a semiconductor produce electron–hole pairs (EHPs), which are separated by a pn-junction or a Schottky diode to generate an electric current. Since the energy of a beta-electron is in the keV range, a single beta-particle creates thousands of EHPs, whereas a photon coming from the Sun typically creates just one pair. On the other hand, the incident photon flux from the Sun is about $10^{6}$ times as high as the flux of the incident betas. As a result, the output power density of a beta-cell is lower than that of a solar cell by at least a factor of a thousand.

Out of all possible sources of beta-particles, tritium has a special place for several reasons. It is environmentally friendly, because it turns into helium-3, a harmless inert gas, and does not emit alpha- and gamma-radiation as a result of a beta-decay. It has a relatively long half-time of 12.3 years. The maximal energy of a beta-particle emanating from tritium is too low to create defects in the crystal lattice of the cell (although indirect radiation damage induced by the X-rays is still possible [8]). Last but not least, tritium is one of the most affordable beta-sources [6].

The energy $E_{EHP}$ to generate a single EHP by a beta-particle is known to scale linearly with the bandgap $E_{g}$ as
(1)$$E_{EHP}=A\;E_{g}+B$$
with A=2.8, B=0.5 eV according to the early work [9] and A=1.54 and B=1.89 eV according to more recent data [10]. Since the highest energy that can possibly be extracted from a single EHP is the bandgap energy, the ultimate efficiency of a beta-battery can be estimated as [6]
(2)$$\eta_{max}=\frac{E_{g}}{AE_{g}+B}\;.$$ This formula suggests that one should use broad-bandgap materials for best efficiency value. For this reason, recent studies have been focused on such materials as diamond [11,12], silicon carbide [13,14,15,16,17], gallium arsenide [18], zinc oxide [19], and gallium nitrate [20]. If one adopts Klein’s parameters [9], one obtains the ultimate efficiency of a beta-battery based on a broad-bandgap semiconductor 1/A=36%; using the parameters from [10], one obtains an even more optimistic estimate of 65%, which, admittedly, is too good to be true.

In this work, we consider a silicon-based betavoltaic element in spite of the fact that silicon has a relatively narrow band gap of 1.12 eV. It is motivated by two considerations. First, silicon is the standard material in semiconductor technologies from microelectronics to solar cells; hence, a silicon-based autonomous power source can be easily combined with other semiconductor devices. Second, due to its wide use in technology, silicon is also the most studied semiconductor material. This means that the theoretical results obtained for a silicon beta-cell will be numerically most accurate within the model assumptions made.

Coming back to the efficiency, the estimate (Equation 2) completely ignores the recombination losses, which can be classified into two groups. The so-called extrinsic recombination channels, such as Shockley-Read-Hall and surface recombination [21], can be controlled by improving the purity of the material and applying surface passivation, whereas the intrinsic radiative and Auger recombination mechanisms cannot be turned off in this way. To obtain a more realistic estimate of the limit efficiency than offered by Equation (Equation 2), one needs to focus on an idealized design of a betavoltaic cell, in which all extrinsic recombination mechanisms are non-operative, so that only the intrinsic ones remain. This approach is standard in the evaluation of the limit efficiency value of solar cells [22,23,24,25].

The important difference between beta- and photovoltaic cells is that a radioisotope-loaded element must be embedded into the material of a beta-battery. For example, one may use tritiated titanium atoms [26], which necessarily will act as Shockley–Read–Hall recombination centers. Perhaps a less detrimental alternative, which is characterized by a lower recombination rate is to tritiate silicon directly, as described in References [27,28]. While it does not seem possible to completely eliminate extrinsic recombination in a beta-battery, it still can be controlled by the choice of technology used to introduce the radioisotope into the battery. In this paper, we address the question of the theoretical limit efficiency of a beta-battery in the limit of a zero extrinsic recombination rate.

The ideal beta-cell design is described in the next section together with the mathematical formalism used for numerical evaluation of the limit efficiency. This formalism is based on the thin-base approximation, which is employed in photovoltaics research [22,23,24,25]. One of the main results of the present paper is that in the case of betavoltaics, this approximation is almost exact, even if the extrinsic recombination channels are included into it by adding the respective term in the current–voltage relation, see Equation (Equation 7) below.

In the limit of a zero extrinsic recombination rate, we show that recombination proceeds predominantly via the Auger mechanism. Based on these findings, analytical expression for the cell current-voltage curve is obtained, from which the betaconversion efficiency is derived. Analytical results are compared with the numerical ones and are shown to be accurate to better than 0.1%. The device efficiency turns out to be a non-monotonic function of the cell thickness, with the optimal thickness of 3 μm being practically independent of the activity of the beta source. The limiting efficiency of a Si beta cell is then found to increase logarithmically with the surface activity $S_{A}$ of the beta-source as $\eta =13.1\;\%+\left(0.449\;\%\right)\;\ln(S_{A}/\left(1\;\mathrm{mCi}/{\mathrm{cm}}^{2}\right))$ within a broad range of $S_{A}$.

## 2. The Model

We focus on the idealized design of a beta-cell shown in Figure 1. A thin layer of tritiated silicon [27,28] is sandwiched between two i-Si slabs of thickness *w* each, where EHPs are produced. The beta-generated EHPs diffuse towards the outer surfaces of each slab, which contain interdigitated heavily doped thin n^+^ and p^+^ regions. There, the EHPs are separated: the holes are collected by the p-electrodes, and the electrons by the n-electrodes. The regions of the same polarity are connected with one another, and the load is applied between the n^+^ and p^+^ groups of electrodes.

Intrinsic rather than doped silicon is chosen as the basis material of the battery in order to minimize the recombination losses. Indeed, it has been shown that the maximal limit efficiency of a silicon solar cell is achieved in the limit of zero doping [23], and same is true in the case of a betavoltaic cell [3]. In this idealized model, we assume that the T/Si layer is so thin that the self-absorption effect can be neglected. We note that the self-absorption can be included by multiplying the efficiency obtained within our idealized model with the ratio of the number of EHPs that enter the semiconductor per unit time to the total activity of the source. According to [28], the maximal beta-activity that one may achieve in this way in tritiated silicon is slightly below 20 mCi/cm^2^.

The device half-thickness *w* should be commensurate with the penetration depth of the electrons from the high-energy part of the beta-spectrum; using the Kanaya–Okayama formula [29], this gives 4.2 μm, see also (Equation 28) below. Both surfaces are covered with an ideally reflecting dielectric coating to prevent photons produced in radiative recombination from escaping the cell; those photons are recycled by the cell.

Excess carrier concentration is governed by the stationary reaction–diffusion equation
(3)$$D\;\frac{d^{2}\Delta n}{dx^{2}}-U\left(\Delta n\right)+G\left(x\right)=0\;,$$
where *D* is the ambipolar diffusion coefficient in i-Si, U(Δn) is the net recombination rate, and G(x) is the rate of electron–hole pair generation by the betas that originate in the yz-plane at x=0. The reaction–diffusion equation is supplemented with the boundary conditions
(4)$$\frac{d\Delta n(x)}{dx}{|}_{x=0}=0\;,\;\;D\frac{d\Delta n(x)}{dx}{|}_{x=w}=-\frac{J}{2q_{e}}\;,$$
where *J* is the current density collected. The first condition follows from the symmetry Δn(−x)=Δn(x). The factor 1/2 in the second condition takes care of the fact that each slab contributes only a half to the net current density *J*.

Instead of the interdigitated configuration, one might use single n^+^ and p^+^ layers deposited on the opposite sides of the battery for current collection. The mathematical treatment of such a seemingly simpler configuration, however, would be somewhat more complicated than Equations (Equation 3) and (Equation 4), because the electric field generated by those electrodes would penetrate the device at not too small voltages. Debye screening length in intrinsic silicon scales with excess carrier concentration as $\lambda_{D}=\sqrt{\epsilon_{0}\epsilon_{Si}kT/\left(2\left(n_{i}+\Delta n\right)\;q_{e}^{2}\right)}∼\left(0.1\;\mu m\right)\sqrt{10^{15}\;{\mathrm{cm}}^{-3}/\Delta n}$, where $\epsilon_{0}$ is vacuum permittivity, $\epsilon_{Si}=11.7$ is the dielectric constant of Si, kT=26meV at room temperature, and $n_{i}$ is the intrinsic concentration. Hence, the Debye length becomes comparable with *w* at Δn of the order of $10^{13}\;{\mathrm{cm}}^{-3}$, which corresponds to a voltage of about V∼0.4V (see Equation (Equation 9) below), which is typical in device operation. Fortunately, in the maximal-power regime, the excess concentration $\Delta n∼10^{15}\;{\mathrm{cm}}^{-3}$, implying that for w∼1−2 μm Equations (Equation 3) and (Equation 4) should be reasonably accurate, as $w\gg \lambda_{D}$. This would necessitate a replacement of our description in terms of the EHP concentration with the one that treats electrons and holes separately. Furthermore, an electric field inside the battery may potentially break the symmetry of the EHP generation by the beta-electrons, and therefore the battery itself would have to be non-symmetrical. This means that instead of a single geometric parameter *w*, we would have to optimize the battery performance with respect to the thicknesses of two slabs, one to the right and the other to the left of the T/Si layer. Both complications are avoided by the use of the interdigitated geometry in Figure 1.

## 3. Thin Base Approximation

Let us focus for now on the simpler case when the recombination rate is a linear function of excess concentration, U(Δn)=Δn/τ, with the recombination time τ independent of Δn. Then, the solution of the reaction–diffusion Equation (Equation 3) reads
(5)$$\Delta n\left(x\right)=C\;\cosh\frac{x}{L}+\sqrt{\frac{\tau }{D}}\int_{-w}^{w}dx^{\prime }\;G\left(x^{\prime }\right)\;e^{-|x-x^{\prime }|/L}\;,$$
where the constant *C* is to be found from the second boundary condition (Equation 4), and where $L=\sqrt{D\tau }$ is the diffusion length, which can be very long (millimeters) in a pure, defect-free silicon sample. Since the generation function G(x) goes very quickly to zero at distances *x* of the order of only a few microns (see Section 4.1 below), it is sensible to take the relevant half-thickness *w* to be in the micrometer range, meaning that the condition w≪L is fulfilled to very high accuracy. For this reason, we can set in (Equation 5) cosh(x/L) and $e^{-|x-x^{\prime }|/L}$ to 1, resulting in the thin-base approximation
(6)Δn(x)=Δn=const. This approximation was originally introduced to analyze the limit efficiency of solar cells [22], whose typical thickness of about 100 μm greatly exceeds the beta-cell thickness 2w. Even in the “thick” solar cells, the accuracy of the thin-base approximation against the full solution of the reaction–diffusion Equation (Equation 3) has been confirmed, see [24].

While the condition w≪L is a mathematical reason to expect that the excess carrier profile in the beta-cell is uniform, there is also a physical reason for this. It has to do with photon recycling. Namely, suppose that Δn(x) is non-uniform; then, photons will be produced in radiative recombination predominantly in the region of high and reabsorbed in the region of low excess carrier concentration, resulting in an overall leveling of the concentration profile [25].

Integrating the reaction–diffusion Equation (Equation 3) from −w to *w* and using the boundary condition (Equation 4) and the constancy of Δn(x), we obtain the net current density as the difference between the beta-generated and recombination currents,
(7)$$J=J_{\beta }-J_{rec}\left(V\right)\;,\;\;J_{rec}\left(V\right)=2wq_{e}U\left(\Delta n\left(V\right)\right)\;,$$
the first main equation from which the J−V curve can be obtained. Here, the beta-generated current density is given by
(8)$$J_{\beta }=q_{e}\int_{-w}^{w}dx\;G\left(x\right)=q_{e}S_{A}N_{\beta }\left(w\right)\;,$$
where $q_{e}$ is the elementary charge, $S_{A}$ is the activity per unit surface area of the beta-source, and $N_{\beta }$ is the mean number of EHPs produced by a single beta-electron in the semiconductor. It is a monotonically increasing function, which saturates at *w* of the order of a few micrometers.

The second equation gives the excess concentration based on the condition $np=n_{i}^{2}\;e^{q_{e}V/kT}$, where $n_{i}$ is the intrinsic concentration. Since in an intrinsic sample the concentrations of electrons and holes are $n=p=n_{i}+\Delta n$, we have
(9)$$\Delta n=n_{i}\left(\Delta n\right)\;\left(e^{q_{e}V/\left(2kT\right)}-1\right)\;.$$ The intrinsic concentration $n_{i}$ weakly depends on Δn due to the bandgap narrowing effect [30],
(10)$$n_{i}\left(\Delta n\right)=n_{i0}\;e^{\Delta E_{g}\left(\Delta n\right)/\left(2kT\right)}\;,$$
where $\Delta E_{g}\left(\Delta n\right)$ is the bandgap narrowing size. Intrinsic concentration in i-Si at Δn=0 was calculated according to [31] with the temperature-dependent bandgap found in [32].

Once Equations (Equation 7) and (Equation 9) are solved numerically, the efficiency of a beta-battery is found as a ratio of the maximal power to the power carried by the betas per unit area,
(11)$$\eta =\frac{J\left(V_{m}\right)V_{m}}{S_{A}E_{\beta }}\;,$$
where $E_{\beta }$ is the average energy of a beta-particle (see (Equation 13) below), and the voltage at maximal power $V_{m}$ is numerically found from the condition ${d\left(J\left(V\right)V\right)/dV|}_{V=V_{m}}=0$. Of interest are also the short-circuit current density $J_{SC}=J_{\beta }$, the open-circuit voltage defined by $J(V_{OC})=0$, and the fill factor of the J−V curve, $FF=J_{m}V_{m}/\left(J_{SC}V_{OC}\right)$.

## 4. Analytical Evaluation of the Betaconversion Parameters

### 4.1. Generation Function

Up to a normalization constant, kinetic energy probability distribution of beta-electrons follows from Fermi’s theory of beta decay and is given by [33]
(12)$$W_{\beta }\left(E\right)=\sqrt{E(E+2m_{e}c^{2})}\;\left(E+m_{e}c^{2}\right)\;{\left(E_{max}-E\right)}^{2}\Theta \left(E_{max}-E\right)\;\frac{2\pi \eta_{S}}{1-e^{-2\pi \eta_{S}}}\;,$$
where $m_{e}c^{2}=511$ keV is electron rest energy, Θ(x) is Heaviside theta, the maximal energy of beta-electrons corrected with respect to the recoil is $E_{max}=18.572$ keV [33], and the Sommerfeld parameter is expressed in terms of the fine structure constant α=1/137.036 and kinetic energy *E* as $\eta_{S}=\alpha Z\left(E+m_{e}c^{2}\right)/\sqrt{E^{2}+2Em_{e}c^{2}}$ with Z=2 being the charge number of the daughter nucleus. With these parameters, the average energy of a beta-electron is found numerically to be
(13)$$E_{\beta }=\frac{\int_{0}^{E_{max}}dE\;E\;W_{\beta }\left(E\right)}{\int_{0}^{E_{max}}dE\;W_{\beta }\left(E\right)}=5.6898\ldots \;\mathrm{keV}\;.$$

Rather than using the phenomenological relation (Equation 2), we took the energy $E_{EHP}=3.67\;\mathrm{eV}$ to generate a single electron–hole pair in silicon directly from the plot in Figure 16 of [10]. In principle, this value should slightly decrease with the excess concentration Δn, because it scales linearly with the bandgap, which becomes narrower with increasing Δn [30]. But since Δn does not exceed $10^{15}\;{\mathrm{cm}}^{-3}$, the size of the bandgap narrowing is not bigger than 1.3 meV. With A≈1.54 [10], this translates into a decrease in $E_{EHP}$ by a mere 2 meV. In other words, for a Si-T battery, the effect of bandgap narrowing on the generation function can be neglected.

Once the energy of a beta-electron drops below a certain threshold value, it becomes unable to generate the EHPs. Setting the threshold energy to 7 eV, we find the number of EHPs produced by a beta-electron in an infinitely thick generation region w→∞
(14)$$N_{sat}=\left(E_{\beta }-7\;\mathrm{eV}\right)/E_{EHP}=1548\;.$$ At finite *w*, the number of EHPs is smaller than this value because of the escape of some beta-particles. Although the threshold energy of 7eV was taken somewhat arbitrarily, its increase by a few tens of eV affects $N_{sat}$ by less than 1%.

A high-energy beta-particle in the material experiences a stopping force, which depends on energy according to an empirical formula [34]
(15)$$F\left(E\right)=-\frac{dE}{ds}=785\;\frac{\rho Z}{AE}\;\ln\left(1.166(kJ+E)/J\right)\;.$$ Here, *E* is measured in eV, F(E) in eV/Å, the distance *s* is measured along the velocity of the beta, Z=14 and A=28.085 are atomic charge and mass numbers, and $\rho =2.329\;g/{\mathrm{cm}}^{3}$ is the density of Si, for which J=172eV, and k=0.822. This formula becomes inaccurate at electron energies below 50 eV [34]. Still, we also used it in the simulations in the range 7 eV <E< 50 eV, as the error introduced by this approximation should be quite small.

In the simulations, the main quantity of interest is the single-particle generation function $g_{\beta }\left(x\right)$, i.e., the average number of EHPs produced by a single beta per unit distance. It is directly related to the generation function from (Equation 3) by
(16)$$G\left(x\right)=S_{A}\;g_{\beta }\left(x\right)\;.$$ To compute $g_{\beta }\left(x\right)$, we slightly modified the procedure explained in [35]. Namely, we simulated $10^{6}$ trajectories of beta-particles originating at x=0. The initial energy of a beta was sampled from the distribution (Equation 12), and the initial angle θ between the particle’s velocity and the *x*-axes had an isotropic distribution
(17)f(θ)=sinθ,θ∈(0,π/2). To simulate a battery consisting of two identical slabs from Figure 1, we applied reflecting boundary conditions at x=0 and absorbing boundary conditions at x=w.

As the beta slows down due to the stopping force (Equation 15), it generates EHPs; the number of EHPs produced in a dx-interval is
(18)$$dg_{\beta }\left(x\right)=F\left(E\right)\;dx/\left(E_{EHP}\;\cos\theta \right)\;.$$ From time to time, a beta particle undergoes elastic collisions with the total cross-section derived by Henoc and Maurice [36] based on Rutherford theory with the incorporation of screening effect:(19)$$\sigma_{R}^{T}\left(E\right)=\left(5.21\cdot 10^{-21}\;{\mathrm{cm}}^{2}\right)\;\frac{Z^{2}}{E^{2}}\;\frac{4\pi }{\alpha (1+\alpha )}\;{\left(\frac{E+mc^{2}}{E+2mc^{2}}\right)}^{2}\;,\;\;\alpha =3.2\cdot 10^{-3}\;Z^{2/3}\;,$$
with *E* measured in eV, Z=14, and $mc^{2}=511\;\mathrm{eV}$. The Mott total cross-section can be found in Reference [37]. The probability for a beta-electron to undergo an elastic collision at a distance *s* from its starting position measured along the beta velocity is governed by
(20)$$\frac{dP(x)}{ds}=-\frac{P(s)}{\lambda (E(s))}$$
where the energy of the beta decreases according to (Equation 15) and the energy-dependent mean free path is related to the total cross-section by
(21)$$\lambda \left(E\right)=\frac{1}{n_{at}\sigma^{T}\left(E\right)}$$
with the concentration of Si atoms $n_{at}=4.994\cdot 10^{28}\;{\mathrm{cm}}^{-3}$.

Suppose that before a collision, a beta-particle was moving in the xy-plane at an angle $\theta_{i}$ relative to the positive *x*-direction, i.e., its direction of motion was specified by a unit vector
(22)$${\hat{u}}_{i}={\hat{e}}_{x}\;\cos\theta_{i}+{\hat{e}}_{y}\;\sin\theta_{i}\;.$$ A collision results in a change of the direction of motion, but not of the energy of the beta. To find the new direction vector ${\hat{u}}_{f}$, we proceeded in three steps. First, the angle of propagation was changed by a random value $\theta_{R}$ found from
(23)$$R={\left(\int_{0}^{\pi }\;d\theta \;\sin\theta \;\frac{d\sigma }{d\Omega }\right)}^{-1}\;\int_{0}^{\theta_{R}}\;d\theta \;\sin\theta \;\frac{d\sigma }{d\Omega }\;,$$
where *R* is a random number uniformly distributed between 0 and 1. For the Rutherford cross-section, the solution of this equation is particularly simple [37]:(24)$$\cos\theta_{R}=1-\frac{2\alpha R}{1+\alpha -R}$$ The new direction vector in the first step becomes
(25)$$\hat{u}={\hat{e}}_{x}\;\cos\left(\theta_{i}+\theta_{R}\right)+{\hat{e}}_{z}\;\sin\left(\theta_{i}+\theta_{R}\right)\;.$$

In the second step, this vector is rotated around ${\hat{u}}_{i}$ by a random angle ϕ uniformly distributed between 0 and 2π. This operation is performed with the help of a rotation matrix R, whose components can be found in section 9.2 of [38]. Taking advantage of the fact that $\hat{u}$ does not have a *z*-component, the rotation matrix is somewhat simpler than the general form:(26)$$\begin{array}{cc} \; & {\hat{u}}_{f}=R\;\hat{u}\;,\\ \; & R=\left(\begin{array}{ccc} \cos\phi +\cos^{2}\theta \;\left(1-\cos\phi \right) & \cos\theta \;\sin\theta \;(1-\cos\phi ) & \sin\theta \;\sin\phi \\ \cos\theta \;\sin\theta \;(1-\cos\phi ) & \cos\phi +\sin^{2}\phi \;\left(1-\cos\theta \right) & -\cos\theta \;\sin\phi \\ -\sin\theta \;\sin\phi & \cos\theta \;\sin\phi & \cos\phi \end{array}\right)\;. \end{array}$$

In the third step, we rotate the coordinate axes so as to have the *z*-component of the new direction vector ${\hat{u}}_{f}$ to be zero. This is achieved by first finding the new angle between the velocity and the *x*-axis,
(27)$$\theta_{f}=\cos^{-1}\frac{u_{fx}}{|u_{f}|}\;,$$
and then redefining
$${\hat{u}}_{f}\to {\hat{e}}_{x}\;\cos\theta_{f}+{\hat{e}}_{y}\;\sin\theta_{f}\;.$$

Both Rutherford (simpler) and Mott (more accurate) scattering cross-section were tried. For both choices, the single-particle generation function $g_{\beta }\left(x\right)$ turned out to be the same; it is shown in Figure 2a for an infinite slab thickness *w*.

It is perhaps somewhat surprising that $g_{\beta }\left(x\right)$ is so robust with respect to a choice of the scattering cross-section. We believe that this robustness has to do with the initial isotropic distribution of the angle θ. This isotropy is not affected by the elastic scattering, no matter which cross-section one uses. Therefore, also the generation function should depend very little on this choice.

The largest distance that a beta-particle can travel while still being able to generate EHPs is
(28)$$w_{max}=\int_{7\;\mathrm{eV}}^{E_{max}}\;\frac{dE}{F(E)}\approx 4.43\;\mu m\;.$$ Actually, the generation function goes to zero even at smaller distances, because the initial direction of motion of a beta is isotropic rather than perpendicular to the yz-plane, and because a beta undergoes multiple elastic scatterings that make its trajectory deviate from a straight line. It is seen in Figure 2a that already after 2 μm from the surface, the generation function drops from the original value by three orders of magnitude. From this, we can conclude that it is sensible to focus on the relatively low values of the element half-thickness *w* between 1 and 2 μm.

At finite *w*-values, we found that numerical results for the average number of electron–hole pairs generated by a single beta can be fitted by a simple expression:(29)$$N_{\beta }\left(w\right)=N_{sat}\left(1-e^{f(w)}\right)$$
with f(w) decreasing almost linearly with *w* within the relevant *w*-range, as found from a cubic fit of $\ln\left(1-N_{\beta }\left(w\right)/N_{sat}\right)$:(30)$$f\left(w\right)=-\frac{w}{w_{0}}+0.07417{\left(\frac{w}{w_{0}}\right)}^{2}-0.007737{\left(\frac{w}{w_{0}}\right)}^{3}\;,\;\;w_{0}=0.2413\;\mu m\;.$$ The high accuracy of this fit is evident from Figure 2b. At $w>w_{max}$, the number of the betagenerated EHPs must be exactly equal to $N_{sat}$, which implies that $f(w>w_{max})$ should be equal to −∞. But the error of the cubic approximation (Equation 30) is negligibly small for all practical purposes.

### 4.2. Recombination Rate

Since we are interested in the limit efficiency of a beta-battery, we assume that the only two recombination channels operative are radiative and Auger recombination, ignoring the Shockley–Read–Hall and surface recombination mechanisms.

The Auger recombination rate in i-Si is
(31)$$U_{A}=C_{A}\left(\Delta n\right){\left(n_{i}+\Delta n\right)}^{3}-C_{A}\left(0\right)n_{i}^{3}\;,$$
with the ambipolar Auger coefficient $C_{A}\approx 2.1\cdot 10^{-30}\;{\mathrm{cm}}^{6}/s$, as reported in [39,40]. Due to the Debye screening by free carriers, the coefficient $C_{A}$ depends on the concentration of electrons and holes via a correction factor of the order of ${\left(\Delta n/N_{ref}\right)}^{2}$ with the reference concentration $N_{ref}=4\cdot 10^{17}\;{\mathrm{cm}}^{-3}$ [40]. Although we included this factor in our numerical calculations, it affects the value of $U_{A}$ by not more than 0.01%, because the highest excess carrier concentration in the open-circuit regime is of the order of $10^{15}\;{\mathrm{cm}}^{-3}$.

The surfaces of the battery are assumed to be covered with a perfectly reflecting dielectric coating, which prevents the photons produced in radiative recombination events from leaving the cell. Those photons may be either reabsorbed by free charge carriers or by valence electrons, which become promoted into the conduction band. The former process is described by the free carrier absorption coefficient $\alpha_{FCA}$, given by an empirical expression from [41]. The radiative recombination rate in an intrinsic semiconductor is, then, given by
(32)$$\begin{array}{ccc} & & U_{r}=B_{r,eff}\left(\Delta n\right)\;{\left(n_{i}\left(\Delta n\right)+\Delta n\right)}^{2}-B_{r,eff}\left(0\right)n_{i0}^{2}\;,\\ & & B_{r,eff}\left(\Delta n\right)=\int_{0}^{\infty }dE\;\alpha_{BB}\left(E\right)\;\frac{n_{r}^{2}\left(E\right)}{c^{2}}\;\frac{8\pi E^{2}}{h^{3}n_{i}^{2}\left(\Delta n\right)}\;e^{-E/kT}\frac{\alpha_{FCA}\left(E,\Delta n\right)}{\alpha_{BB}\left(E\right)+\alpha_{FCA}\left(E,\Delta n\right)}\;. \end{array}$$ Here, $n_{i}\left(\Delta n\right)$ is given by (Equation 10), $\alpha_{BB}\left(E\right)$ is the band-to-band absorption coefficient, $n_{r}\left(E\right)$ is the refractive index of silicon [42], and *h* is Planck’s constant. The term that multiplies the ratio $\alpha_{FCA}/\left(\alpha_{BB}+\alpha_{FCA}\right)$ in the effective recombination coefficient $B_{r,eff}$ follows from Würfel’s generalization of Planck’s radiation law [43], while this ratio signifies the fraction of photons that are lost due to reabsorption by free carriers.

### 4.3. Analytical Approximation of the Current-Voltage Curve

#### 4.3.1. Recombination Current

In most radiative recombination events, photons with energy $E\approx E_{g}+kT$ are produced. At such energies, the band-to-band absorption coefficient $\alpha_{BB}$ is of the order of $1\;{\mathrm{cm}}^{-1}$ [42]. On the other hand, the free carrier absorption coefficient can be estimated as $\alpha_{FCA}\approx \left(10^{-17}\;{\mathrm{cm}}^{2}\right)\left(n_{i}+\Delta n\right)$, as follows from Equation (Equation 7) of [41]. Hence, the ratio of the absorption coefficients from (Equation 32) can be estimated as
(33)$$\frac{\alpha_{FCA}}{\alpha_{BB}+\alpha_{FCA}}\approx \left(10^{-17}\;{\mathrm{cm}}^{3}\right)\left(n_{i}+\Delta n\right)\;.$$ Even at the highest relevant Δn of the order of $10^{15}\;{\mathrm{cm}}^{-3}$, this ratio is only about $10^{-2}$. Taking it outside of the integral in (Equation 32) and performing integration numerically at T=298.15K, we find that the radiative recombination rate depends on excess carrier concentration in a way similar to Auger recombination rate, namely,
(34)$$U_{r}\approx \left(5\cdot 10^{-32}\;{\mathrm{cm}}^{6}\right)\;\Delta n^{3}$$
at $\Delta n\gg n_{i}$. Since the prefactor of $5\cdot 10^{-32}\;{\mathrm{cm}}^{6}$ is about the same as the discrepancy between the values of $C_{A}$ reported in [39,40], viz. $2.11\cdot 10^{-30}$ and $2.06\cdot 10^{-30}\;{\mathrm{cm}}^{6}$, radiative recombination in a beta-cell from Figure 1 proceeds much slower than Auger recombination.

With this in mind, we approximate the recombination current in (Equation 7) with a simpler expression that involves the Auger channel only:(35)$${\tilde{J}}_{rec}\left(V\right)=2wq_{e}C_{A0}\;n_{i0}^{3}{\left(e^{q_{e}V/\left(2kT\right)}-1\right)}^{3}\;.$$ Here, the Δn-dependence of Auger recombination constant and intrinsic concentration are neglected; they are both taken at V=0 and Δn=0. We further neglected the terms related to the intrinsic concentration in the Auger recombination rate (Equation 31).

To compensate for the effects that are not included in ${\tilde{J}}_{rec}$, we introduce a fixed parameter K≈1 in the expression for the current density, which now becomes
(36)$$J\left(V\right)=J_{\beta }-K{\tilde{J}}_{rec}\left(V\right)\;.$$ The constant *K* is chosen so as to correctly reproduce the J−V curve in the vicinity of some reference voltage $V_{r}$, which is sensible to choose close to the open-circuit value, i.e., from the condition ${\tilde{J}}_{rec}\left(V_{r}\right)=J_{\beta }$. This gives
(37)$$V_{r}=\frac{2kT}{q_{e}}\;\ln\left({\left(\frac{S_{A}N_{\beta }}{2wC_{A0}n_{i0}^{3}}\right)}^{1/3}+1\right)\;,\;\;K=\frac{J_{rec}\left(V_{r}\right)}{{\tilde{J}}_{rec}\left(V_{r}\right)}\;.$$ For the parameter range considered here (*w* between 0.1 and 100 μm and $C_{A}$ between 0.1 and 100 mCi), the constant *K* varied by only a few percent around the value K=1.13.

Figure 3 demonstrates the high accuracy of the approximate J−V relation (Equation 36) with K=1.13 (solid line) as compared to the numerical results (symbols). Even taking the simplest value K=1 (dashed lines), the discrepancy between analytical and numerical results is not too poor, as the approximate analytical curve turns out to be horizontally shifted relative to the numerical counterpart by only about 2 mV. With K=1, the accuracy of all relevant betaconversion parameters would be about 0.5%, whereas with K=1.13, the betaconversion parameters discussed below differ from the numerically precise values only in the fourth or fifth significant figure.

#### 4.3.2. Betaconversion Parameters

By setting the current (Equation 36) to zero, we obtain the open-circuit voltage
(38)$$V_{OC}=\frac{2kT}{q_{e}}\;\ln\left({\left(X/K\right)}^{1/3}+1\right)\;,$$
where we used the expression (Equation 8) for $J_{\beta }$ and introduced the variable
(39)$$X=\frac{S_{A}N_{\beta }\left(w\right)}{2wC_{A0}n_{i0}^{3}}\;.$$

Focusing on the practically relevant case of the maximal-power voltage notably exceeding the thermal voltage, $V_{m}\gg kT/q_{e}$, we neglect 1 in the brackets of (Equation 35) and find the voltage at maximal power by setting the derivative of $P\left(V\right)=J\left(V\right)V=J_{\beta }V-2Kwq_{e}C_{A0}n_{i0}^{3}e^{3q_{e}V/\left(2kT\right)}V$ to zero. The result reads
(40)$$V_{m}=\frac{2kT}{3q_{e}}\left(W_{L}\left(e^{1}X/K\right)-1\right)\;,$$
where Lambert’s function is defined by the condition $W_{L}\left(x\right)\;e^{W_{L}\left(x\right)}=x$; an efficient numerical algorithm to compute it is presented in [44].

Substituting (Equation 39) into the current density expression and using the definition of $W_{L}\left(x\right)$, we find
(41)$$J_{m}=q_{e}S_{A}N_{\beta }\left(w\right)\;\left(1-\frac{1}{W_{L}\left(e^{1}X/K\right)}\right)\;.$$ Finally, the efficiency of a beta-battery is given by
(42)$$\eta =\frac{2kTN_{\beta }\left(w\right)}{3E_{\beta }}\frac{{\left(W_{L}\left(e^{1}X/K\right)-1\right)}^{2}}{W_{L}\left(e^{1}X/K\right)}\;.$$

An interesting prediction of these formulae is that the voltages $V_{OC}$ and $V_{m}$, as well as the ratio $\eta E_{\beta }/\left(N_{\beta }\left(w\right)kT\right)$, do not depend on the source surface activity $S_{A}$ and cell half-thickness *w* separately, but rather on their combination (Equation 39). This is indeed confirmed in Figure 4, which shows the numerical results of Equations (Equation 7)–(Equation 9), where the approximation-free recombination rate is given by the sum of Equations (Equation 31) and (Equation 32) (symbols), and analytical formulae (Equation 38), (Equation 40) and (Equation 42). The agreement between the two sets of data is obvious.

### 4.4. Optimal Thickness and Maximal Efficiency

Shown in Figure 5a is the dependence of the betaconversion efficiency on cell half-thickness at three different values of surface activity $S_{A}=0.1,1$, and 10 mCi/cm^2^. It is seen that these curves are non-monotonic and develop a maximum at around $w_{opt}=1.5\;\mu$m. The origin of this non-monotonicity is easy to understand. Initially, increasing the half-thickness results in a larger number of beta-electrons that are able to deposit all of their energy in the cell rather than fly out of the cell. As the thickness increases above the value (Equation 28), the total number of EHPs produced per unit time does not grow with *w* anymore; however, the excess concentration of the EHPs decreases with *w*, resulting in a reduction of efficiency.

Setting the derivative of η(w) with respect to *w* to zero, we obtain after some algebra an equation that determines the optimal half-thickness, at which the efficiency is maximized:(43)$$W_{L}\left(\frac{e^{1}}{2KC_{A0}n_{i0}^{3}}\;\frac{N_{\beta }\left(w_{opt}\right)}{w_{opt}}\;S_{A}\right)=g\left(w_{opt}\right)\;,\;\;g\left(w\right)=\frac{1-e^{-f(w)}}{f^{\prime }\left(w\right)w}\;.$$ An analytical solution of this transcendental equation for $w_{opt}$ is not attempted here. Instead, we use the definition of Lambert’s *W*-function to express surface activity of the beta source at which a particular optimal thickness $w_{opt}$ is realized:(44)$$S_{A}=\frac{2KC_{A0}n_{i0}^{3}w_{opt}}{e^{1}N_{\beta }\left(w_{opt}\right)}\;g\left(w_{opt}\right)\;e^{g(w_{opt})}\;\;.$$ One can use $w_{opt}$ as a parameter, whose substitution into (Equation 44) gives the corresponding surface activity. Substitution of the so-obtained $S_{A}$ and $w_{opt}$ into (Equation 42) gives the maximal efficiency $\eta_{max}$. The results of these calculations are shown in Figure 5b. It is seen that the optimal thickness $w_{opt}$ varies extremely weakly with the surface activity, changing by less than 5% as $S_{A}$ changes by three orders of magnitude.

The maximal efficiency increases logarithmically with $S_{A}$. Fitting the curve from Figure 5b gives
(45)$$\eta =13.1\;\%+\left(0.449\;\%\right)\;\ln\frac{S_{A}}{1\;{\mathrm{mCi}/\mathrm{cm}}^{2}}\;.$$ In fact, to obtain the $\eta_{max}$ vs. $S_{A}$ dependence, one may take a constant value of $w_{opt}$ around 1.5 μm; the resulting values of $\eta_{max}$ will change at most in the fourth significant figure.

## 5. Conclusions

Silicon is an interesting candidate material for betavoltaic applications not only due to its wide availability, but also due to its long lifetimes of excess charge carriers that can be achieved by producing pure, defect-free samples. In this work, we discussed the performance of a silicon-based beta-cell coupled to a tritium source of beta particles. In this design, the tritiated silicon layer is sandwiched between two intrinsic Si slabs of equal thickness, and interdigitated p- and n-type regions are produced on the two surfaces of the battery for current collection. The choice of i-Si as the base region material is motivated by its lower recombination rate as compared to the p- or n-doped Si, and the interdigitated electrode geometry is preferred in order to eliminate the electric field that would exist in the battery if simple planar p- and n-electrodes were deposited on its opposite sides.

To calculate the limit betaconversion efficiency, we neglected the effect of the parasitic shunt and series resistance and the extrinsic recombination mechanisms. Out of the only two remaining recombination channels are radiative and Auger recombination, the former is effectively turned off by coating the device surface with a reflective layer, which prevents the photons from escaping the cell. Those photons are reabsorbed by the semiconductor, typically with the generation of a new electron–hole pair; only a small part of photons is lost due to the free charge carrier absorption.

The advantage of the thin-base approximation used in this work lies in the fact that it does not rely on a solution of a differential equation for the position-dependent excess concentration profile, which is assumed to be uniform. Because of the small size of a beta-battery and due to the effect of photon reabsorption, the thin-base approximation is an adequate tool to analyze betavoltaic power sources. In fact, we expect that it can also be used to analyze the efficiency of the beta-batteries based on other semiconductor materials with a beta source in the presence of the extrinsic Shockley–Read–Hall, surface, and space–charge region recombination mechanisms. When only intrinsic (radiative and Auger) recombination is operative, this analysis is greatly simplified by the fact that the radiative recombination rate in the presence of photon recycling scales with excess concentration in the same way as the Auger recombination rate, allowing one to combine the two contributions to the recombination current into a single term and making it possible to develop an accurate analytical approximation scheme.

The efficiency of this beta-battery depends non-monotonically on the cell thickness and develops a maximum at the optimal thickness $2w_{opt}=3\;\mu$m, practically independent of the surface activity of the beta-source. The maximal efficiency of the cell, on the other hand, increases logarithmically with source activity from about 12.1% to 14.1% as activity is raised from 0.1 to 10 mCi/cm^2^.

This has the following implication with respect to a beta-cell performance. As the beta-source becomes depleted over the years, its activity $S_{A}$ decreases exponentially. Simultaneously, the efficiency of the cell $\eta \propto \ln S_{A}$ will be decreasing linearly in time, and hence the output power of the cell, $P_{out}\propto S_{A}\;\eta \left(S_{A}\right)$, is expected to decrease faster than exponentially. 

## Figures and Tables

**Figure 1 micromachines-14-02015-f001:**
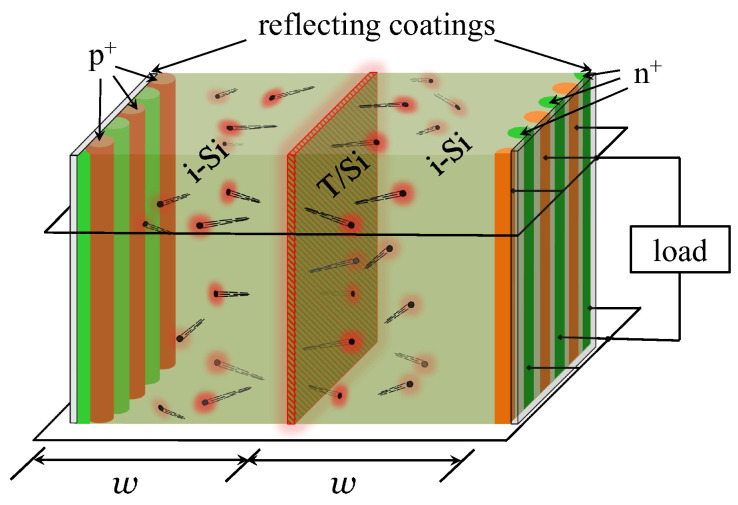
Schematics of a beta-battery consisting of two i-Si slabs of thickness *w* each separated by a thin layer of tritiated silicon. The current is collected by the heavily doped n^+^ (green) and p^+^ (orange) interdigitated electrodes. The electrodes of the same polarity are connected with one another, and the load is applied between the electrodes of opposite polarities. The outer surface of the battery is covered with a reflecting coating that prevents photons produced in radiative recombination from escaping the device; these photons are reabsorbed and recycled.

**Figure 2 micromachines-14-02015-f002:**
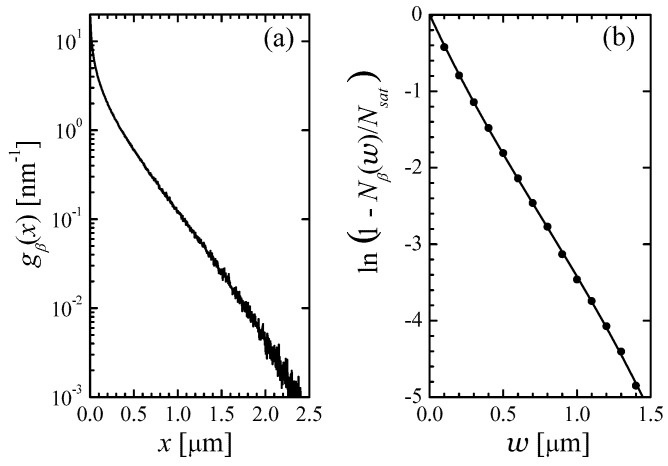
(**a**) Single -particle generation function vs. distance from the beta-source. (**b**) The combination $\ln\left(1-N_{\beta }\left(w\right)/N_{sat}\right)$ vs. cell half-thickness obtained in simulations (circles) and the function f(w) from (Equation 30) (solid line).

**Figure 3 micromachines-14-02015-f003:**
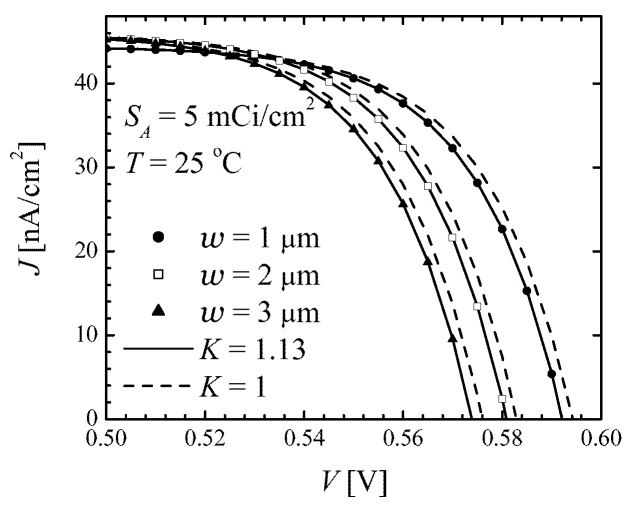
Current–voltage curve of an ideal Si/T beta-battery with the source activity $S_{A}=5\;\mathrm{mCi}/{\mathrm{cm}}^{2}$ at *T* = 25 °C for three values of cell half-thickness, as indicated in the legend. Symbols: numerical solution of Equations (Equation 7) and (Equation 9) with the net recombination rate given by the sum of Auger (Equation 31) and radiative (Equation 32) contributions. Solid lines: approximate analytical expression (Equation 36) with K=1.13. Dashed lines: (Equation 36) with K=1.

**Figure 4 micromachines-14-02015-f004:**
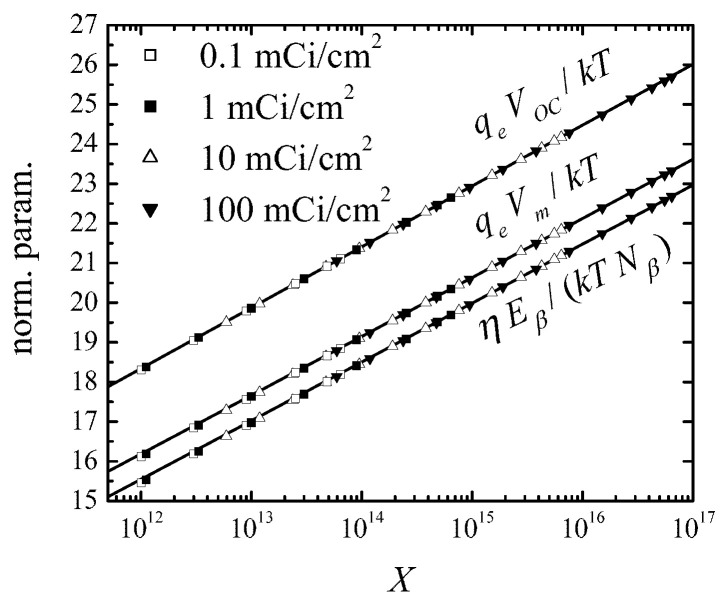
Open-circuit voltage and voltage at maximal power normalized to thermal voltage, $kT/q_{e}$, and betaconversion efficiency normalized to the ratio $kTN_{\beta }\left(w\right)/E_{\beta }$, as functions of the parameter *X* from (Equation 39). Solid lines: analytical formulae (Equation 38), (Equation 40) and (Equation 42) with K=1.13; symbols: numerical results obtained for surface activity values $S_{A}=0.1,1,10$, and 100 mCi/cm^2^. When generating the numerical data, the half-thickness *w* of the cell varied between 0.1 and 500μm.

**Figure 5 micromachines-14-02015-f005:**
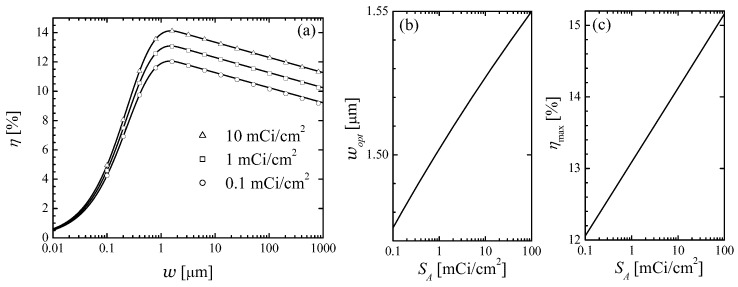
(**a**) Betaconversion efficiency of an ideal cell vs. cell half-thickness for three values of the surface activity of beta-source. Symbols: numerical results; lines: analytical approximation. (**b**) Optimal half-thickness that maximizes cell efficiency vs. surface activity of the beta-source, obtained using (Equation 44). (**c**) Maximal cell efficiency vs. surface activity of the beta source.

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
