# Peer review of "Limit Efficiency of a Silicon Betavoltaic Battery with Tritium Source"

_micromachines, 2023, doi:10.3390/mi14112015_

Round 1

Reviewer 1 Report

Comments and Suggestions for Authors

The structure proposed in the author's article is novel, taking into account photon cycling effects and designing a high-efficiency silicon-based nuclear cell, so I would recommend acceptance after minor revision.

Question 1: On page 2, line 49, the author explains that silicon is the most mature semiconductor with abundant theoretical models, but on page 2, line 52, the author ignores the Shockley-Read-Hall composite model in the process of building the model. the Shockley-Read-Hall composite model is one of the main mechanisms of non-radiative composites, and the theoretical simulation of silicon semiconductor is relatively mature, and the deuterated titanium is prepared with high efficiency, so I recommend the minor revision to accept it. The theoretical simulation of silicon semiconductors is relatively mature, and deuterated titanium must have deep defect energy levels during the preparation process, the effect of Shockley-Read-Hall composite is not negligible, and there are many theoretical simulation models, please explain why this composite mechanism was ignored.

Question 2: On page 4, line 112, the author describes "defect-free silicon sample", and on page 12, line 346, the author states that "pure, defect-free samples" can be produced. "This is not true. The formation of P-type or N-type materials from intrinsic silicon semiconductors must be carried out by means of doping. By doping a pure silicon semiconductor with a donor impurity, the semiconductor becomes an N-type semiconductor, for example, by doping silicon with phosphorus, and the phosphorus forms a donor energy level, which is essentially a defect energy level. Silicon without defects cannot form a PN junction. The author has misrepresented himself in these points, which are not in line with reality or theory.

Question 3: Please discuss the amount of tritium stored in silicon tritium storage material, which will affect the activity of the radioactive source in the nuclear cell, through Eq. 45 which in turn will be able to give the theoretical maximum energy conversion efficiency.

Question 4: Please discuss the distribution of energy deposition of the radioactive isotope tritium in the structure, adding to the discussion of the self-absorption effect of the radioactive source to show the structural advantage.

Question 5: Please reorganize the conclusion section to highlight and discuss the structural advantages and explain why the article is designed in such a way as to achieve limit efficiency.

Author Response

Dear Editors,

We would like to thank both Referees for their promptness, constructive suggestions, and especially for recommending our paper for publication in Micromachines. Please find below our response to the Referees’ comments. To address the Referees’ comments, we added a few sentences to the manuscript. The newly added text is indicated in red. We had to slightly modify the surrounding sentences to ensure a smooth transition between the already present and the newly added text; those modifications are cosmetic in nature.

Best regards,

Mykhaylo Evstigneev

Referee 1

Question 1: On page 2, line 49, the author explains that silicon is the most mature semiconductor with abundant theoretical models, but on page 2, line 52, the author ignores the Shockley-Read-Hall composite model in the process of building the model. the Shockley-Read-Hall composite model is one of the main mechanisms of non-radiative composites, and the theoretical simulation of silicon semiconductor is relatively mature, and the deuterated titanium is prepared with high efficiency, so I recommend the minor revision to accept it. The theoretical simulation of silicon semiconductors is relatively mature, and deuterated titanium must have deep defect energy levels during the preparation process, the effect of Shockley-Read-Hall composite is not negligible, and there are many theoretical simulation models, please explain why this composite mechanism was ignored.

Authors’ response. Good point. It is explaned in the two new paragraphs that we added in Section 1.

Question 2: On page 4, line 112, the author describes "defect-free silicon sample", and on page 12, line 346, the author states that "pure, defect-free samples" can be produced. "This is not true. The formation of P-type or N-type materials from intrinsic silicon semiconductors must be carried out by means of doping. By doping a pure silicon semiconductor with a donor impurity, the semiconductor becomes an N-type semiconductor, for example, by doping silicon with phosphorus, and the phosphorus forms a donor energy level, which is essentially a defect energy level. Silicon without defects cannot form a PN junction. The author has misrepresented himself in these points, which are not in line with reality or theory.

Authors’ response. The p- and n-type doping is applied only on the cell surface, where interdigitated regions are created for current collection, while the electron-hole pairs are indeed produced in a defect-free intrinsic region. We have added more detail on that in the beginning of Section 2.

Question 3: Please discuss the amount of tritium stored in silicon tritium storage material, which will affect the activity of the radioactive source in the nuclear cell, through Eq. 45 which in turn will be able to give the theoretical maximum energy conversion efficiency.

Authors’ response: This piece of information is now also added in the beginning of Section 2.

Question 4: Please discuss the distribution of energy deposition of the radioactive isotope tritium in the structure, adding to the discussion of the self-absorption effect of the radioactive source to show the structural advantage.

Authors’ response: The distribution of energy deposition is discussed in Section 4.1. The effect of self-absorption is neglected in our idealized model. We emphasized this in a newly added text in the beginning of Section 2.

Question 5: Please reorganize the conclusion section to highlight and discuss the structural advantages and explain why the article is designed in such a way as to achieve limit efficiency.

Authors’ response: Done.

Reviewer 2 Report

Comments and Suggestions for Authors

An idealized and new design of a silicon betavoltaic battery with a tritium source is presented. The calculated efficiency is very high for Si-based betavoltaics, which is interesting for readers in similar field. I recommend its publication in Micromachine after major revision. 

1. In introduction part, some recent progress on betavoltaics should be introducted, as follow:

1)" Quantitative modeling, optimization, and verification of 63Ni-powered betavoltaic cells based on three-dimensional ZnO nanorod arrays",  NUCL SCI TECH (2022) 33:144. (0123456789().,-volV)https://doi.org/10.1007/s41365-022-01127-6 ;

2)" Betavoltaic Nuclear Battery: A Review of Recent Progress and Challenges as an Alternative Energy Source ",J. Phys. Chem. C 2023, 127, 16, 7565–7579.

2. The description of work princple of betavoltaics raised by authors was not clear, please provide a schematic diagram to exexplain the generation, separation and transport of beta-excited EHPs. 

3. The energy depositon and penatration thickness of beta particles in i-Si should be simulated and calculated for designing a optimum thickness of Si slab.

Author Response

Dear Editors,

We would like to thank both Referees for their promptness, constructive suggestions, and especially for recommending our paper for publication in Micromachines. Please find below our response to the Referees’ comments. To address the Referees’ comments, we added a few sentences to the manuscript. The newly added text is indicated in red. We had to slightly modify the surrounding sentences to ensure a smooth transition between the already present and the newly added text; those modifications are cosmetic in nature.

Best regards,

Mykhaylo Evstigneev

Referee 2

1. In introduction part, some recent progress on betavoltaics should be introducted, as follow:

1)" Quantitative modeling, optimization, and verification of 63Ni-powered betavoltaic cells based on three-dimensional ZnO nanorod arrays",  NUCL SCI TECH (2022) 33:144. (0123456789().,-volV)https://doi.org/10.1007/s41365-022-01127-6 ;

2)" Betavoltaic Nuclear Battery: A Review of Recent Progress and Challenges as an Alternative Energy Source ",J. Phys. Chem. C 2023, 127, 16, 7565–7579.

Authors’ response: These references are now added and cited in the new version.

2. The description of work princple of betavoltaics raised by authors was not clear, please provide a schematic diagram to exexplain the generation, separation and transport of beta-excited EHPs. 

Authors’ response: This is now done in the first paragraph of Section 2.

3. The energy depositon and penatration thickness of beta particles in i-Si should be simulated and calculated for designing a optimum thickness of Si slab.

Authors’ response: This is done in Section 4.1 of the manuscript.

Round 2

Reviewer 2 Report

Comments and Suggestions for Authors

The authors have addressed all my questions, the revised manuscript can be accepted in present form.